# New Deep Learning Model to Estimate Ozone Concentrations Found Worrying Exposure Level over Eastern China

**DOI:** 10.3390/ijerph19127186

**Published:** 2022-06-11

**Authors:** Sichen Wang, Xi Mu, Peng Jiang, Yanfeng Huo, Li Zhu, Zhiqiang Zhu, Yanlan Wu

**Affiliations:** 1School of Resources and Environmental Engineering, Anhui University, Hefei 230601, China; x19301096@stu.ahu.edu.cn (S.W.); x21201036@stu.ahu.edu.cn (L.Z.); zhuzq@imagesky.com.cn (Z.Z.); 12059@ahu.edu.cn (Y.W.); 2Information Materials and Intelligent Sensing Laboratory of Anhui Province, Hefei 230601, China; x20301066@stu.ahu.edu.cn; 3Anhui Province Engineering Laboratory for Mine Ecological Remediation, Anhui University, Hefei 230601, China; 4Anhui Institute of Meteorological Sciences, Hefei 230031, China; huoyanfeng@ahmi.org.cn

**Keywords:** ozone pollution, deep learning, human exposure, eastern China, tropospheric monitoring instrument, long short-term memory network, data-driven model

## Abstract

Ozone (O_3_), whose concentrations have been increasing in eastern China recently, plays a key role in human health, biodiversity, and climate change. Accurate information about the spatiotemporal distribution of O_3_ is crucial for human exposure studies. We developed a deep learning model based on a long short-term memory (LSTM) network to estimate the daily maximum 8 h average (MDA8) O_3_ across eastern China in 2020. The proposed model combines LSTM with an attentional mechanism and residual connection structure. The model employed total O_3_ column product from the Tropospheric Monitoring Instrument, meteorological data, and other covariates as inputs. Then, the estimates from our model were compared with real observations of the China air quality monitoring network. The results indicated that our model performed better than other traditional models, such as the random forest model and deep neural network. The sample-based cross-validation R^2^ and RMSE of our model were 0.94 and 10.64 μg m^−3^, respectively. Based on the O_3_ distribution over eastern China derived from the model, we found that people in this region suffered from excessive O_3_ exposure. Approximately 81% of the population in eastern China was exposed to MDA8 O_3_ > 100 μg m^−3^ for more than 150 days in 2020.

## 1. Introduction

In the stratosphere, naturally occurring ozone (O_3_) protects organisms from the harmful solar ultraviolet radiation [1]. However, ground-level O_3_ is harmful to humans and other organisms at high concentrations [2]. Integrated findings from toxicological animal studies and epidemiological studies allow us to state that O_3_ can react with unsaturated fatty acids, amino groups, and other proteins, causing chest pain, memory loss, and decreased vision [3]. In recent years, although many emission control measures have been stringently enforced, China has experienced severe O_3_ pollution [4]. During 2015–2020, the O_3_-related health impacts for all-cause and respiratory diseases vastly increased in China [5]. O_3_ pollution was particularly severe in eastern China; the annual daily maximum 8 h average (MDA8) O_3_ in this region for 2015 was the highest nationwide [6], and increased by 13.3% from 2015 to 2017 [7]. In Anhui Province, which lies in the western reaches of the eastern China region, the annual mean O_3_ concentration increased by 15.6% from 2016 to 2020, and the government declared that O_3_ has become the primary air pollutant [8]. Thus, a more efficient method is needed to estimate the spatiotemporal distribution of O_3_ and assess its population exposure level.

Although China has established more than 1600 air quality monitoring sites, most of the sites generally cover limited representative areas [9]. Conventionally, chemical transport models (CTMs) and statistical models have been developed to compensate for the inadequate spatiotemporal coverage of monitoring sites. The main features of CTMs and statistical models have been described in many reports in the literature. In sum, CTMs (e.g., WRF-Chem, GEOS-Chem) simulate the environmental processes of O_3_ and its precursors with the support of meteorological data and emission inventories [10,11]; their results tend to be more interpretable, but the CTMs generally show higher uncertainties because of imperfect understanding of O_3_-related chemical mechanisms. Statistical models (e.g., land-use regression (LUR), machine learning, deep learning) build regression equations between predictive variables and O_3_. Although pure mathematical analyses lack the in-depth understanding of chemical mechanisms, statistical models can achieve higher accuracies with less computational requirements. For example, recently some CTM studies for China O_3_ estimation reported that the coefficient of determination values were 0.5~0.7 [12,13], while those of statistical models could be greater than 0.8 [14,15]. Therefore, statistical models are considered a reasonable means of providing high-quality data for air pollution control.

The LUR model is one of the most widely used statistical models, which uses land use types, meteorological elements, and other geographic factors near the monitoring site as model inputs to build models for estimation of atmospheric pollutant concentrations where there are no sites [16]. Although LUR has developed into a general model to simulate the spatial distribution of city-scale atmospheric pollutants, it uses linear regression for parameter fitting. Considering the complex and nonlinear relationship between O_3_ and its predictors (e.g., surface pressure, solar radiation), LUR is not an optimal option for modeling O_3_. Machine learning is another widely used technique based on statistical learning theory. Efficient nonlinear algorithms enable ML models to capture the complex relationships between the response and predictors [17]. A comparison between seven ML models and five LUR models across the United States showed that the ML models were superior in estimating the MDA8 O_3_ concentrations [18]. Recently, several machine learning algorithms, such as support vector regression [19], random forest [20] and extreme gradient boosting [21] have been successfully applied to map the distributions of air pollutants. Although most researchers reported comparatively stronger performance, these models have relatively low complexity and lack accurate calculation of the spatiotemporal heterogeneities of air pollutants.

Deep learning techniques have some more sophisticated algorithms for processing the potential spatiotemporal relationships between variables. Long short-term memory (LSTM) network is a typical sequential network, which takes time-series data as inputs and is recurrent along the time dimension. Using parameter sharing, LSTM can remember information at different time steps [22]. Compared to classical machine learning algorithms mentioned above, LSTM can infer the complex effects of variables over a time period on the estimated results. For example, LSTM can not only learn the effects of today’s solar radiation on ozone formation, but also can determine the potential role of solar radiation in the past. In the field of atmospheric pollution research, LSTM has been widely used in the prediction of changes in site observations in advance. For example, Kim et al. used LSTM to predict daily PM_10_ and PM_2.5_ in South Korea, reported better performance with the LSTM than with the 3-D CTM simulations [23]. Chang et al. developed an LSTM model for PM_2.5_ prediction in Taiwan and reported that LSTM has obvious advantages over support vector machine regression and gradient boosted tree regression [24]. Comparative research on O_3_ prediction in Malaysia also reported that LSTM performed better than some ordinary machine learning algorithms [25]. Although these works have great significance for pollution control, the full-coverage O_3_ spatial distribution can not be obtained as a result of the built models based on site observations.

The main challenges in the application of LSTM in satellite-based spatial predictions of O_3_ over a large area can be summarized as follows: (1) The O_3_ concentration is very dynamic, especially in the troposphere, but the temporal resolutions of remote sensing observations are relatively low. For example, most trace-gas-monitoring satellites are in a sun-synchronous orbit, and can only provide daily products. (2) The quality control of satellite-based/reanalysis data is not as robust as in situ observations; complicated physical retrieval or assimilation processes can result in several uncertainties. To solve these problems, we embedded an attention mechanism in the LSTM model and adopted advanced atmospheric composition observation satellite Sentinel-5 Precursor total O_3_ column retrievals as one of the predictive variables. The model performances including overall fitting ability, spatiotemporal extrapolation, and peak estimation ability were fully evaluated by using observations from the China air pollution monitoring network. The spatiotemporal distribution of MDA8 O_3_ was estimated in 2020 over eastern China at a spatial resolution of 0.1° × 0.1°. O_3_ spatiotemporal patterns and human exposure intensities were determined based on a complete and credible dataset. This work can provide higher quality pollution data products for environmental and epidemiological research.

## 2. Materials and Methods

### 2.1. Study Area and Ground-Level O_3_ Observation

The study area (Figure 1) included four provinces (Shandong, Anhui, Jiangsu, and Zhejiang) and one province-level municipality (Shanghai), located in eastern China, ranging from 26.98° to 38.45° N and 113.96° to 124.04° E. The total land area of the study area was 517,000 km^2^, with a population of 327 million, accounting for approximately 5.2% and 23% of China’s land area and total population, respectively. It is thus one of the most populated zones in the world. In addition, the industrial level of the region is highly developed, with one of China’s largest comprehensive industrial centers (Shanghai-Nanjing-Hangzhou Industrial Base), and its GDP in 2020 was 31% of China’s total GDP.

The hourly O_3_ concentrations for 2020 were obtained from the website of the China National Environmental Monitoring Center (CNEMC) [26]. The arrangements of monitoring sites followed HJ 664-2013 specifications [27]. UV spectrophotometry was employed to measure the O_3_ concentrations, and HJ 818-2018 specifications were used to ensure the data quality [28].

### 2.2. Tropospheric Monitoring Instrument Total O_3_ Column

Many studies have reported acceptable accuracy and reasonable consistency between satellite-retrieved total O_3_ columns and surface O_3_ [29,30] concentrations. Additionally, the useful ozone monitoring instrument (OMI)’s total O_3_ column in machine learning surface O_3_ modeling has been reported. However, OMI has too many invalid data records due to the sensor’s physical obstruction, which largely limits the application of OMI. As a successor of the ozone monitoring instrument (OMI), the tropospheric monitoring instrument (TROPOMI) on board the Sentinel-5P (S5P) satellite aims to observe global atmospheric components. Compared to OMI, the spatial resolution of TROPOMI is significantly higher, with a signal-to-noise ratio improving between 1–5 times [31]. Therefore, various trace gases such as O_3_, SO_2_, NO_2_, and CO can be measured more accurately. The total O_3_ column Level-2 data product (5.5 × 3.5 km) was obtained from Copernicus Open Access Hub [32]. The TROPOMI-O_3_ was retrieved using a direct-fitting algorithm, and the bias with respect to the ground-based O_3_ column density measurements was 3.5–5% [33].

### 2.3. Meteorological Data and Other Covariates

Meteorological variables were obtained from the ERA5 datasets (0.25° × 0.25°) of the European Center for Medium-Range Weather Forecasts [34], including surface solar radiation downwards (SSRD), 2 m dew point temperature (D2M), 2 m temperature (T2M), 10 m eastward wind component (U10), 10 m northward wind component (V10), boundary layer height (BLH), mean sea level pressure (MSL), surface pressure (SP), and total precipitation (TP). The selection of meteorological variables was based on prior knowledge of O_3_. For example, SSRD can reflect the intensity of sunlight, which affects the photochemical reaction of O_3_ precursor pollutants [35]. BLH affects the diffusion space of pollutants in a vertical direction [36]. Wind usually disperses concentrations from the emission sources [37]. Temperature has an effect on the process of precursor generation [38], and precipitation hints unstable atmospheric conditions [39].

Normalized difference vegetation index (NDVI), road density (RD), surface classification (SC), surface elevation (DEM), latitude (LAT), longitude (LON) and day of year (DOY) were also used as predictors. NDVI was obtained from MODIS 16-day product [40], and RD was obtained from Open Street [41]. The SC and DEM were the input data for the retrieval of O_3_ column density and provided together with the TROPOMI-O_3_ product. The TROPOMI SC data were derived from the U.S. Geological Survey Global Land Cover Characterization dataset [42], and DEM were derived from ECMWF and GMTED2010 [43]. In addition, the population counts data (30 arc-second) for 2020 were retrieved from Gridded Population of the World dataset v4 [44].

### 2.4. Date Preprocessing

A small proportion of missing values do not significantly change the spatiotemporal characteristics of data for large-scale and long-term studies, and the most widely used method involves omitting the missing sections. However, considering the limited time and space in this study, there was a possibility of a large amount of missing data in specific spatiotemporal domains that may contain significant information. For O_3_ monitoring sites, the missing values occupied 0.75% to 22.9%. We compared five imputing methods: latest-valid-observation (Baseline), linear interpolation (Linear), cubic spline interpolation (CUBIC), Bayesian probabilistic matrix factorization (BPMF) [45] and low-rank matrix completion (LMC) [46]. Some sites with high data-missing rates (>8%) were identified, and then 20 testing sites with a missing rate of less than 3% near these sites were selected. The root mean square error (RMSE) and mean absolute percentage error (MAPE) were used as the metrics for evaluation of different imputed results.

For TROPOMI data, a quality threshold (ranging from 0 (poor) to 1 (excellent)) is provided with the total O_3_ column product together to remove invalid data caused by poor observation conditions (e.g., cloud cover) and bad retrieved results. Following the official instructions, we only adopted data with a quality threshold greater than 0.5, leading to a few areas with missing data (2.8%). Considering the missing data did not show clear distribution characteristics; we used ordinary kriging (OK) method to process TROPOMI data.

After processing the missing values, we selected the maximal hourly 8 h moving average from 8:00 to 24:00 local time as daily MDA8 ozone level. All predictive variables were resampled to 0.1° × 0.1° using appropriate algorithms. The meteorological and TROPOMI-O_3_ data were resampled using the OK method. The surface classification was resampled using majority resampling. Then, the TROPOMI-O_3_ and meteorological data were extracted at the locations of the ground-based O_3_ monitoring sites, 6 days of data for each variable were fused into a 2-D matrix with a size of 6 × 15 (6 days and 15 variables), and the Z-score normalization method was applied to convert the distribution of original variables into a standard normal distribution.

### 2.5. Model Development

Because the applications of LSTM are extensive and the principles are complex, the schematic formulas will not be repeated in this study. We used three groups of trainable parameter matrices to calculate the attention weight of each unit of the LSTM model. The formulas are as follows:(1)$$score\left(h_{i},h_{t}\right)=Softmax\left(V\left(W_{1}+W_{2}h_{t}\right)\right)$$
(2)$$s_{i}=score*h_{i}$$
(3)$$s_{t}=\sum_{i}^{t}s_{i}$$
where $h_{i}$ denotes the output of LSTM at time i; V and W are the trainable parameter matrices; $s_{i}$ is the output at time *i*; and $s_{t}$ is the final output.

The attention mechanism assigns a weight to the output of each time step, and then the model uses this weight to obtain the final output. First, the attention score at each time step was generated using Equation (1), where the Softmax function was used to convert the calculation result into a probability. Then, the output of each time step was updated via the attentional weight obtained using Equation (2). The final output ($s_{t}$) was the summation of the output at each time step, obtained using Equation (3). In addition, residual connection was adopted to solve the degradation problem in the deep learning model. A one-layer LSTM with the residual connection and attention mechanism (AR-LSTM) is shown in Figure 2.

The parameters of the AR-LSTM model were determined by using exploratory experiments. The preprocessed data were divided into two parts randomly, the training set and testing set (8:2). For a deep learning model, the number of hidden layers, the number of neurons in each hidden layer, the loss function, and optimization should be determined primarily. The grid search technique can be used to provide optimized parameter configuration, but it is almost impossible because of the huge computational cost. Therefore, we set the following parameters: optimizer (Adam), loss function (MSE), activation function (tanh) and initial model structure (hidden layers and a fully connected layer with one neuron), which are commonly used [47]. The number of hidden layers was selected from 1 to 6, and the number of neurons in each hidden layer was determined from 16, 32, 64, 128, and 256. The time steps were selected from 1 to 15. In addition, the automatic decay of the learning rate and early stopping techniques were applied to reduce over-fitting.

### 2.6. Model Evaluation

Four cross-validation (CV) methods were used for model validation. The most frequently used CV methods are sample-based CV and site-based CV. The sample-based CV method is appropriate for evaluating the overall performance of the model, whereas the site-based CV method can evaluate the spatial variations in model performance. However, the distribution of sites was not inhomogeneous. There may be some training sites near the testing sites. In this case, the features of the testing sites were closely related to those of the neighboring training sites. Therefore, we introduced a city-based CV method to ensure spatial independence of the testing sites. In the city-based CV method, the data are divided into 10 folds by cities instead of sites. For temporal extrapolation capability, a month-based method was adopted, which left one month out for CV.

The metrics included R^2^, MAE, root mean square error (RMSE), and mean absolute percentage error (MAPE). In addition, peak validation was adopted to judge whether the model could provide a reliable peak concentration estimation. Hit rate (HR), false alarm ratio (FAR), missing rate (MR) and threat score (TS) were used as metrics to evaluate the model ability for estimating MDA8 O_3_ over 100 μg m^−3^, which is the air quality guideline (AQG) recommended by the World Health Organization (WHO). The formulas are as shown below:(4)$$H_{R}=\frac{a}{a+c}$$
(5)$$F_{AR}=\frac{b}{a+b}$$
(6)$$M_{R}=\frac{c}{a+c}$$
(7)$$T_{S}=\frac{a}{a+b+c}$$
where a denotes the number of samples in which observations and predictions are all greater than 100 μg m^−3^; b denotes predictions more than 100 μg m^−3^ but observations are not; c denotes observations more than 100 μg m^−3^ but predictions are not.

### 2.7. O_3_ Level and Human Exposure Assessments

For a given region and period, the population-weighted MDA8 O_3_ concentrations were regarded as the O_3_ level and were calculated using the formulation formula [21]:(8)$$C_{pw}=\overset{N}{\underset{i=1}{\sum }}\left(P_{i}\times C_{i}\right)/\overset{N}{\underset{i=1}{\sum }}\left(P_{i}\right)$$
where $C_{pw}$ is the population-weighted MDA8 O_3_ for the region (N grid cells); $P_{i}$ is the population density for grid i; and $C_{i}$ is the MDA8 O_3_ concentration in grid i.

The nonattainment day was defined as >100 μg m^−3^ based on AQG. The exposure intensities and durations were the cumulative percent of populations exposed to different levels of MDA8 O_3_ [48].

## 3. Results

### 3.1. Missing Data Imputation Results

The results from the five methods for site observations imputation are shown in Figure 3. There was a negative correlation between the performances of imputed methods and the duration of missing data, and the LMC method performed with relative stability. As the missing data of TROPOMI-O_3_ were very small, the OK method had strong applicability. Figure 4 shows the imputation results for TROPOMI-O_3_ data on 11 June; the missing data imputed by the OK method were smooth and agreed well with the surrounding original data.

### 3.2. Model Configuration Selections

The performance of the AR-LSTM model with different parameters is shown in Figure 5. The number of neurons directly affected model fitting ability, but too many neurons would increase the computational cost and lead to over-fitting. We found that the model performed optimally with the number of neurons equaling 128 (Figure 5h). Stacked multi-layers can improve the nonlinear capability of deep learning models. As shown in Figure 5f, the model performance gradually stabilized with the number of hidden layers over four. The time steps determine the amount of information contained in time-series data; the mean R^2^ value remained around 0.92 when the time steps were greater than 6 days (Figure 5g). Therefore, the final AR-LSTM model had four hidden layers, 128 neurons in each hidden layer, and the input time-series data had 6 time steps.

### 3.3. Model Performance and Grid-Data Generation

Figure 6 shows four CV results. The sample-based CV and site-based CV showed similar results, with an R^2^ value of 0.94. The city-based CV showed a significant reduction in R^2^, RMSE, and MAPE values (0.85, 17.25 μg m^−3^, and 16.9%, respectively). The sample divisions in the city-based CV are provided in Appendix A. Like the city-based CV, the results of the month-based CV were also not as good as the first two commonly used CV methods. The R^2^, RMSE, and MAPE values were 0.77, 21.03 μg m^−3^ and 21.17%, respectively. The results suggest that the spatiotemporal extrapolation of AR-LSTM was not as good as its overall prediction ability, and reveal the weak robustness of data-driven algorithms. However, the accuracy of the AR-LSTM model was still acceptable.

The optimal model in sample-based CV was selected for the generation of O_3_ grid-data. The comparison of results showed that there was a high degree of consistency between the in situ observations and generated data, with the R^2^, RMSE, and MAPE equal to 0.94, 10.95 μg m^−3^, and 10.2%, respectively (Figure 7). We selected Tracking Air Pollution China (TAP) MDA8 O_3_ dataset, which was generated by using CTMs and the data fusion method [49], for consistency contrast. Figure 8 shows the spatial patterns of our data were generally in good agreement with the TAP data, and our RMSE was lower. The R^2^ value for each site and month are shown in Appendix A. Spatially, the quality of generated data was not as good in the southwest. This could be because of the sparse distribution of monitoring sites in the south. Temporally, the highest R^2^ values appeared in summer (0.97), and the lowest value was observed in winter (0.90). From the point of view of peak validation, the annual averages of HR, FAR, MR and TS were 0.94, 0.05, 0.06 and 0.90, respectively. We also calculated peak validation metrics of each site (Appendix A) and month (Appendix A), and there was no significant spatial nonstationarity.

### 3.4. Comparisons with Other Methods

We selected several widely used machine learning and deep learning models including RF, deep neural network (DNN), gate recurrent unit (GRU) [50], original LSTM and CNN [51] for comparisons. The key difference between these models is how the spatiotemporal relationship of variables is processed. RF and DNN have no special spatiotemporal relationship calculation logic, GRU and LSTM are two types of recurrent neural network, and CNN uses spatial convolution structure to process spatial information. As for the parameter selection for these models, GRU and LSTM adopted the same parameters as AR-LSTM because the only difference between them was the structure of the internal calculation unit. The parameters of RF, DNN and CNN were determined by using an exploratory approach similar to that of AR-LSTM; the specific process can be found in Appendix A.

Since the purpose of this article was the reconstruction of historical pollution data, the sample-based CV and city-based CV were adopted as validation methods. Table 1 shows the CV results of various models. The performances of LSTM and GRU were better than those of RF and DNN. The results indicated that the time series of variables provided valuable information for O_3_ modeling. However, the LSTM and GRU models processed the time series of variables in a relatively simple manner compared with our AR-LSTM model, leading to less accurate results. For a more detailed explanation, the attention weights of some of the samples are illustrated in Appendix A. We found that the largest attention weights did not appear in the last epoch in approximately 8% of the samples. The assumption of a decrease in the relationship between model inputs and O_3_ with time was not sufficiently accurate. On the other hand, the performances of the CNN models had no clear advantages, and the results indicate that the spatial information was of limited usefulness in this paper. Presumably, that was due to the coarse spatial resolution of the input variables.

In addition, the performance of AR-LSTM was compared with similar studies published recently (Appendix A). Generally speaking, our model reached a relatively high level in term of statistical metrics. However, due to the different study areas and periods, the results are for reference only.

### 3.5. Sensitivity Analysis of Modeling Variables

We carried out sensitivity analysis in the modeling stage. We implemented a variable screening method based on the variable importance of the random forest (RF) model. RF is an ensemble decision tree model, which performs a split according to a given splitting criterion, and the variable importance can be obtained by weighing the improvements in the splitting criterion in all the nodes where the variable appears as a splitter. Therefore, the RF model is more explanatory than other machine learning models and is a natural model for variable sensitivity analysis [52,53]. We built an RF model and removed variables in ascending order based on variable importance; sample-based CV10 was used to evaluate the model performance after removing one variable each time. Appendix A shows the screening results. The model achieved optimal performance after removing road density (RD), probably because of its low temporal resolution. However, the result was not significant enough to support a definitive conclusion. Considering that adding a variable had little effect on the computational burden, all collected data were used in this paper.

### 3.6. Human Exposure Assessment

According to our predictions, the annual average population-weighted MDA8 O_3_ was estimated to be 102.9 ± 33.9 μg m^−3^ across eastern China in 2020 (Table 2), showing an increase of ~13% in five years [54]. Seasonally, the population-weighted MDA8 O_3_ was predicted to be 120.1 ± 33.0, 118.6 ± 19.1, 101.3 ± 33.9, and 67.3 ± 18.4 μg m^−3^ for spring, summer, fall, and winter, respectively. The O_3_ level peaked in spring over all provinces except Shandong. Regionally, the population-weighted MDA8 O_3_ was found to be the highest in Shandong (110.0 ± 43.5 μg m^−3^) and lowest in Zhejiang (94.8 ± 33.3 μg m^−3^). The overall spatial pattern was higher in the north and lower in the south. To some extent, the spatial pattern was related to the regional industrial distribution. Energy-intensive industries, such as petrochemicals and non-ferrous smelting, which produce higher precursor emissions, are the dominant industries in Shandong, but the economy of Zhejiang is dependent on the information technology and tourism industries. In addition, the increase in VOC due to anthropogenic emissions, such as coal combustion in Zhejiang, could be lower than that in Shandong due to the lower population density.

Note that although there were large intra-annual and spatial variations in O_3_ levels in eastern China, the general trends showed severe O_3_ pollution. This was true even in Zhejiang, which showed the best air quality, where the O_3_ levels in all seasons (except winter) exceeded the AQG. Additionally, the summer O_3_ level in Shandong (142.3 ± 36.5 μg m^−3^) was closer to the interim target 1 (IT-1:160 μg m^−3^) formulated by the WHO. Figure 9 shows the exposure intensities and durations, which provide more detailed information about the severity of pollution. Approximately 81% of the population in eastern China lived in areas with more than 150 non-attainment days (i.e., population-weighted MDA8 O_3_ > 100 μg m^−3^) in 2020, and 15% of the population was exposed to O_3_ levels higher than the IT-1 value for more than 60 days.

## 4. Conclusions

This study proposed a new deep learning model to estimate the MDA8 O_3_ concentrations across eastern China on a 0.1° × 0.1° grid. Attention mechanism and residual connection were introduced into each LSTM unit to improve its ability for temporal information processing. With the support of a small number of variables from the public dataset, the AR-LSTM model achieved good performance (sample-based CV R^2^ = 0.94 and city-based CV R^2^ = 0.85). In addition, this model is not limited in O_3_ estimation; it is easy to employ similar model structures for the estimation of other types of air pollution, such as PM_2.5_ or NO_2_. The assessments indicate that eastern China suffered severe O_3_ pollution in 2020. The annual average population-weighted MDA8 O_3_ concentration was estimated to be 102.8 ± 33.1 μg m^−3^, and 81% of the eastern China population lived in areas with more than 150 non-attainment days. The rapid implementation of emergency control measures for O_3_ pollution is essential. Normalized control measures for VOC emissions should be implemented in earnest, especially in Shandong province. The petrochemical industry and companies that use large amounts of organic solvents, photo-oxidized technology, and activated carbon adsorption cotton must be monitored vigorously. The government should actively urge these companies to upgrade or replace their substandard VOC treatment devices. Stricter measures should also be considered, such as banning asphalt and paint spraying operations during the daytime and regulating the number of vehicles based on the O_3_ concentration.

There are some limitations of this study that should be noted. First, the spatiotemporal domain was relatively small. The dataset only contained a one-year temporal series and five provinces. For a data-driven model, the amount of data is one of the key factors affecting model performance. Second, the input variables were not very comprehensive. Regardless of model complexity, more covariates such as emission inventories and other pollutants should be considered, which may improve model accuracy. Finally, TROPOMI-O_3_ was derived from a complex physical retrievals process that may introduce more uncertainties. Estimating surface O_3_ concentrations directly using the origin observations of satellite-based spectral sensors should be considered.

## Figures and Tables

**Figure 1 ijerph-19-07186-f001:**
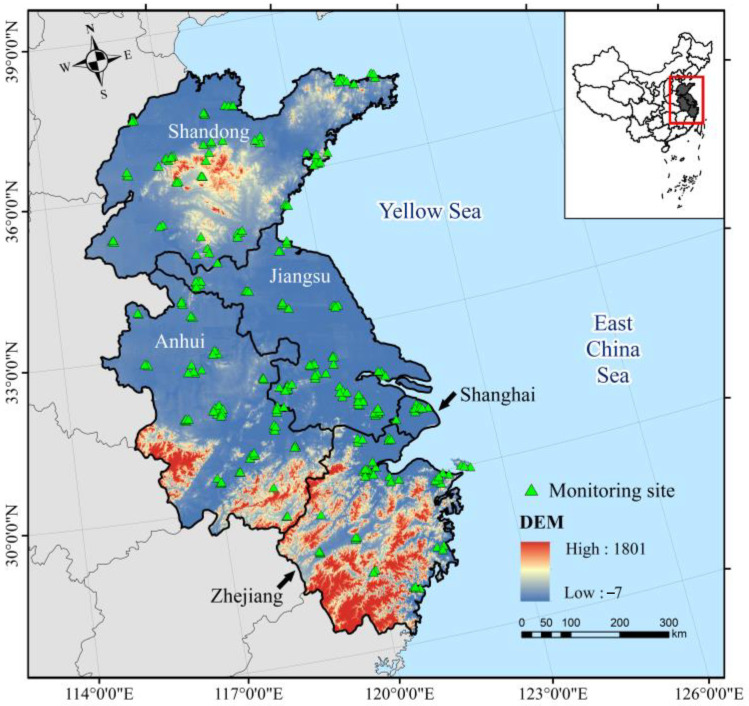
Study area and the distribution of ground O_3_ monitoring sites.

**Figure 2 ijerph-19-07186-f002:**
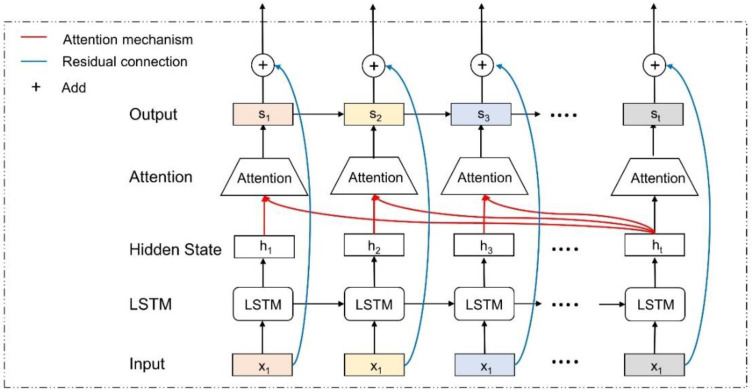
One layer AR-LSTM.

**Figure 3 ijerph-19-07186-f003:**
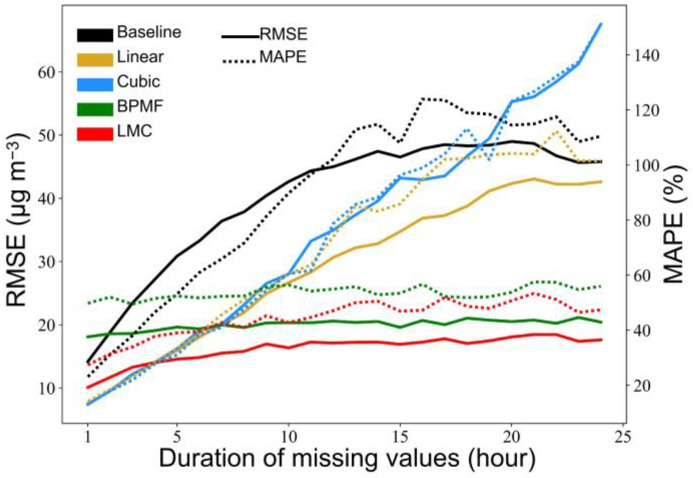
Comparisons of the five imputed methods for in situ missing observations.

**Figure 4 ijerph-19-07186-f004:**
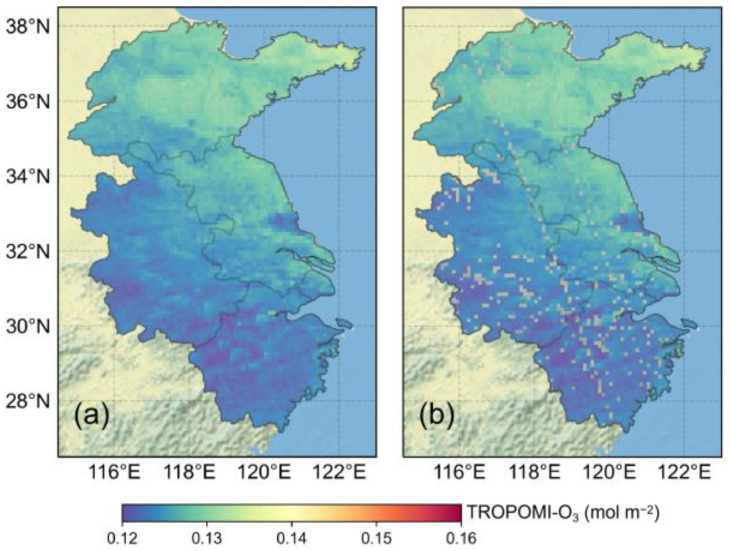
Imputed TROPOMI-O_3_ by using OK method on 11 June (**a**) and original data (**b**).

**Figure 5 ijerph-19-07186-f005:**
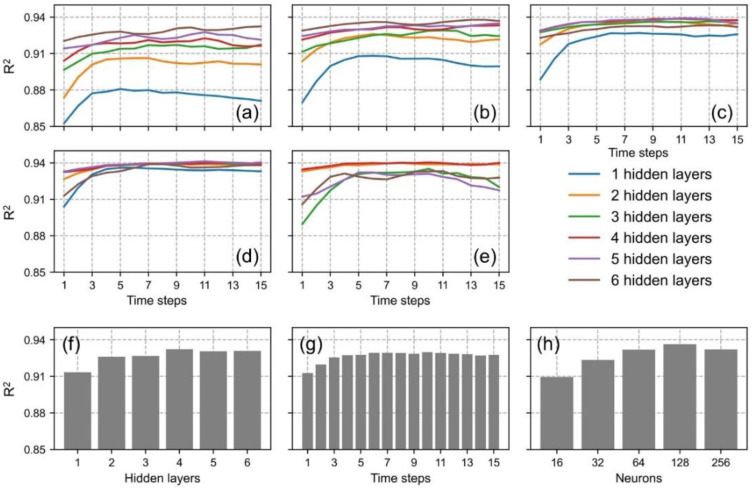
AR-LSTM model performance with different parameters. (**a**–**e**) are the R^2^ curves of 16, 32, 64, 128, and 256 neurons with different layers and time steps, respectively. (**f**–**h**) are mean values of different hidden layers, time steps and different numbers of neurons, respectively.

**Figure 6 ijerph-19-07186-f006:**
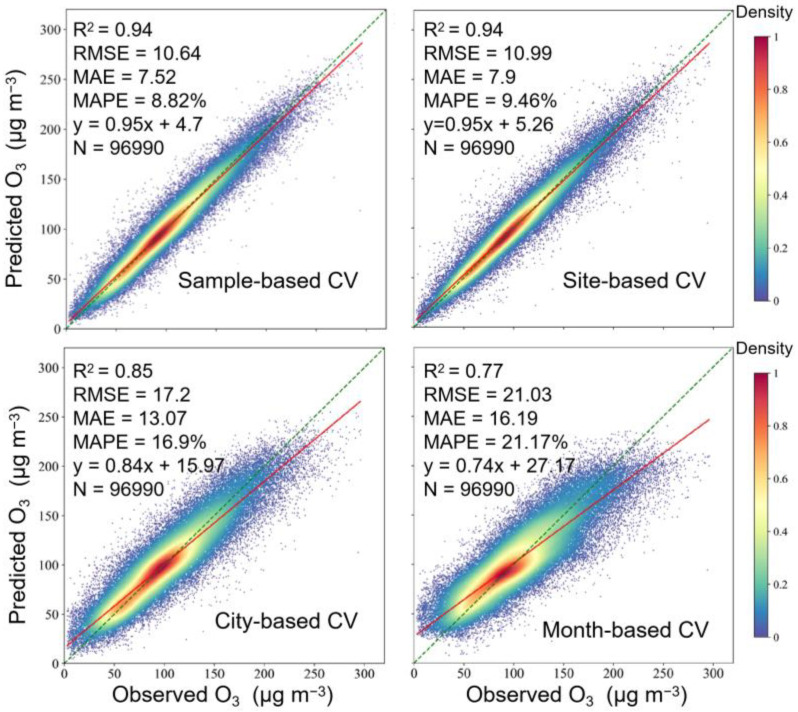
CV results.

**Figure 7 ijerph-19-07186-f007:**
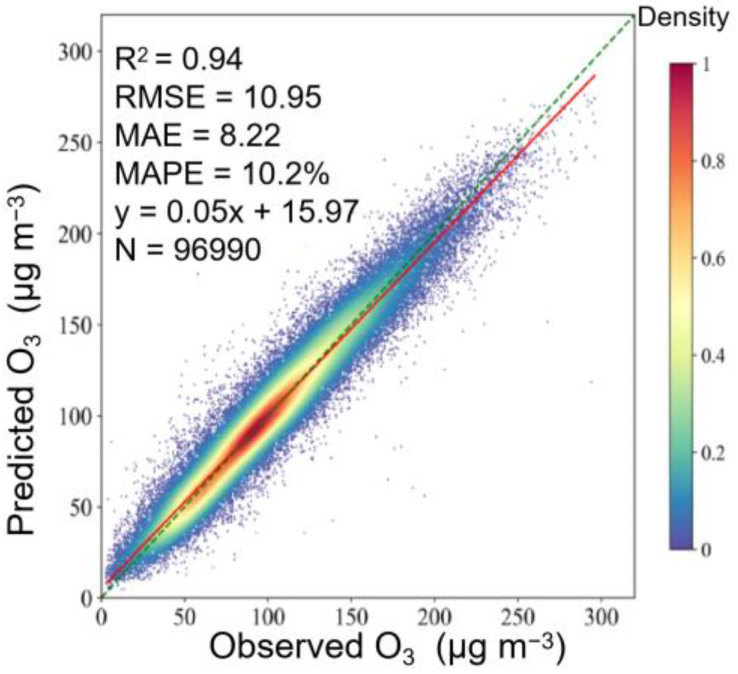
Scatter plot of the generated data and in situ observations.

**Figure 8 ijerph-19-07186-f008:**
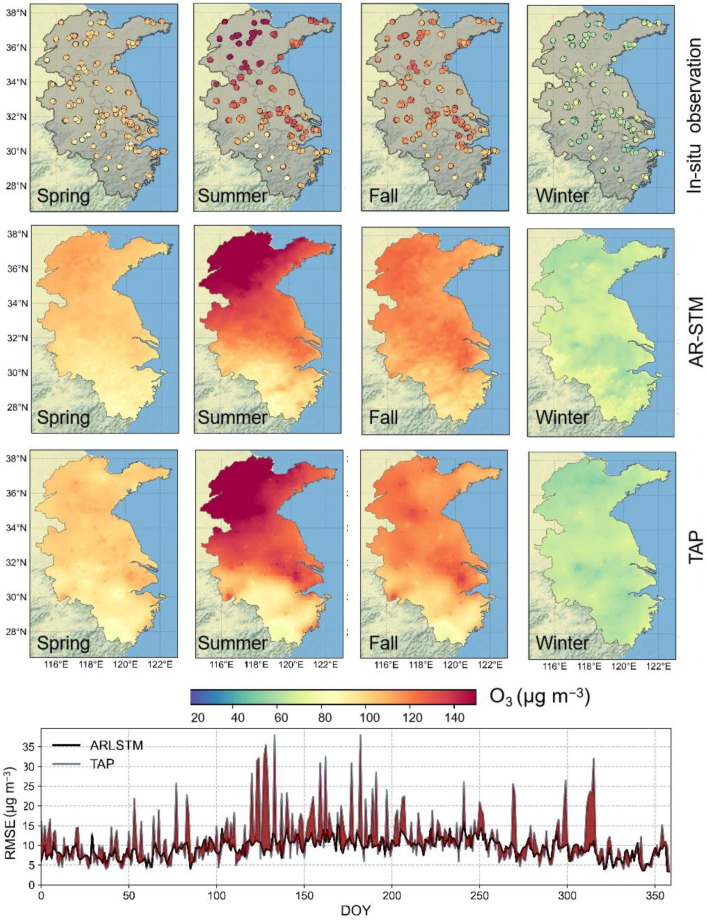
Quarterly averages of MDA8 ozone of TAP, ARLSTM and in situ observations. The line plot below is a comparison of RMSE between the two datasets and in situ observations.

**Figure 9 ijerph-19-07186-f009:**
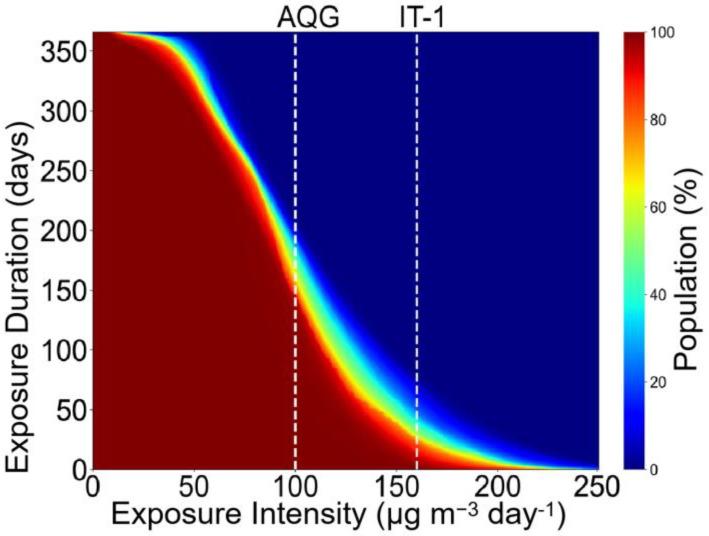
Cumulative distributions of daily mean exposure to O_3_. The two white dotted line are air quality guideline (AQG: 100 μg m^−3^) and interim target 1 (IT1: 160 μg m^−3^), respectively.

**Table 1 ijerph-19-07186-t001:** Performance of various models in sample-based CV and city-based CV. The bold is used to emphasize the best results.

Model	Sample-Based CV				City-Based CV			
	R^2^	RMSE	MAE	MAPE	R^2^	RMSE	MAE	MAPE
RF	0.89	14.33	10.41	13.17	0.79	19.17	14.76	19.45
DNN	0.88	15.28	11.48	14.52	0.79	19.64	14.83	19.80
GRU	0.91	13.28	9.64	11.80	0.80	19.36	14.45	18.61
LSTM	0.92	12.80	9.34	11.45	0.82	18.65	14.10	18.14
CNN	0.90	13.72	10.26	12.96	0.80	19.70	14.92	19.93
AR-LSTM	0.94	10.64	7.52	8.82	0.85	17.25	13.07	16.90

**Table 2 ijerph-19-07186-t002:** Seasonal and annual averages ± deviations of population-weighted MDA8 O_3_ for five provinces and the all of eastern China in 2020 (μg m^−3^).

Region	Spring	Summer	Fall	Winter	Annual
Shandong	124.23 ± 33.14	142.31 ± 36.48	101.99 ± 41.02	66.73 ± 19.14	110.03 ± 43.46
Anhui	116.58 ± 30.77	109.03 ± 23.31	102.24 ± 36.32	67.12 ± 19.44	99.69 ± 33.70
Jiangsu	122.91 ± 32.43	118.40 ± 29.46	102.05 ± 36.33	67.83 ± 19.55	103.87 ± 36.72
Shanghai	119.45 ± 32.50	107.32 ± 39.28	97.71 ± 34.38	70.80 ± 21.41	99.81 ± 36.94
Zhejiang	113.29 ± 33.06	95.53 ± 20.64	100.30 ± 35.72	66.53 ± 23.12	94.76 ± 33.32
Eastern China	120.06 ± 27.98	118.55 ± 19.10	101.33 ± 33.93	67.31 ± 18.36	102.89 ± 32.86

## Data Availability

Not applicable.

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
