# Peer review of "New Deep Learning Model to Estimate Ozone Concentrations Found Worrying Exposure Level over Eastern China"

_ijerph, 2022, doi:10.3390/ijerph19127186_

Round 1
Reviewer 1 Report
In this paper, it proposed a new deep learning model to estimate the O3 concentrations across eastern China on a 0.1° × 0.1° grid. The results showed that the AR-LSTM model achieved good performance. The paper is scientifically sound and should obtain broad international interest on air pollution prediction.
My suggestions for revision are as follows:
1. Meteorological conditions have a great impact on ozone generation. In this paper, the meteorological parameters were obtained from the ERA5 datasets (0.25°×0.25°) of the European Center for Medium-Range Weather Forecasts. The spatial resolutions of the meteorological parameters were different from that of ozone (on a 0.1° × 0.1° grid). In deep learning model, how to coordinate the spatial resolutions of the meteorological parameters and ozone? This needs to be clarified.
2. The author claims that the model has good prediction ability. However, there is almost no physicochemical mechanism related to ozone formation in the AR-LSTM model. Therefore, it is possible that the neural network prediction result is only the optimized fitting result by the parameters adjustment. How to explain the universality of the model? The authors should strengthen the content of discussion.
3. In 2020, China had imposed stringent policies to slow down the spread of coronavirus disease 2019 (COVID-19), closing businesses and factories, restricting travel, and issuing stay-at-home orders. It resulted in air pollutants emissions have also fallen sharply. However, there are almost no ozone precursor’s emissions data related to ozone formation in the AR-LSTM model. Thus, how to ensure the reliability of simulation results? The authors should strengthen the content of discussion.
4. Different data sources have different precision. For the deep learning model, how do different input variables with different data sources in input layers affect prediction results? It lacks the sensitivity analysis of the model.
5. The works should be discussed by comparing the performance with other similar O3 deep learning prediction models.
Author Response
We greatly appreciate you for your high evaluations on this manuscript. Your constructive suggestions are helpful for further improving the quality of this work.
Please see attached file for point-to-point replies.

Reviewer 2 Report
Dear Authors
The manuscript is well written and presents relevant research that deserves to be published in the IJERPH journal.
There are a few recommendations for additions and changes that I list below:
1) The title is too generic. I recommend replacing it with a "newspaper headline" style title that explains to readers very directly what the paper is about. I suggest a similar title like this one: "New deep learning model to estimate Ozone concentrations found an increase in pollution over eastern China";
2) Add more keywords. This will make your paper more visible to search engines like Google Scholar. IJERPH allows up to 10 keywords, so it is advisable that you include 6 more keywords in the manuscript;
3) Lines 29-30: include at least one sentence and a bibliographic reference briefly explaining the impacts on human health and wildlife;
4) The results presented in lines 337-344 and 360-363 are very worrying, from a socio-environmental perspective. The authors are right in recommending the implementation of emergency control measures for O3 pollution. However, the authors do not mention what those measures would be. Please include them in the manuscript.
Greetings from Brazil.
Author Response

(The authors gave the same response as above.)

Reviewer 3 Report
General comments:
This study introduces a new deep learning-based method to estimate the daily maximum 8-hour averaged O3 across Eastern China for the year 2020. The accuracy of the new method compared to the other ML-based methods is promising. And the final science-based conclusions are significant. Despite some strong opinions/statements that are provided in the article without sufficient backing, the article/work is excellent.
A nicely written article. Provides solid background to emphasize the societal importance of the study.
Please see attached file for line-by-line comments.

Author Response

(The authors gave the same response as above.)
